# Human Milk Oligosaccharides: Potential Applications in COVID-19

**DOI:** 10.3390/biomedicines10020346

**Published:** 2022-02-01

**Authors:** Somchai Chutipongtanate, Ardythe L. Morrow, David S. Newburg

**Affiliations:** 1Pediatric Translational Research Unit, Department of Pediatrics, Faculty of Medicine Ramathibodi Hospital, Mahidol University, Bangkok 10400, Thailand; chutipsi@ucmail.uc.edu; 2Department of Clinical Epidemiology and Biostatistics, Faculty of Medicine Ramathibodi Hospital, Mahidol University, Bangkok 10400, Thailand; 3Faculty of Medicine Ramathibodi Hospital, Chakri Naruebodindra Medical Institute, Mahidol University, Samut Prakan 10540, Thailand; 4Division of Epidemiology, Department of Environmental and Public Health Sciences, University of Cincinnati College of Medicine, Cincinnati, OH 45267, USA; morrowa@ucmail.uc.edu; 5Division of Infectious Diseases, Department of Pediatrics, Cincinnati Children′s Hospital Medical Center, University of Cincinnati College of Medicine, Cincinnati, OH 45267, USA

**Keywords:** HMOS, immunomodulation, long-COVID, mucosal signaling, prebiotics, receptor binding inhibition, SARS-CoV-2

## Abstract

Coronavirus disease 2019 (COVID-19) pandemic caused by severe acute respiratory syndrome-coronavirus-2 (SARS-CoV-2) has become a global health crisis with more than four million deaths worldwide. A substantial number of COVID-19 survivors continue suffering from long-COVID syndrome, a long-term complication exhibiting chronic inflammation and gut dysbiosis. Much effort is being expended to improve therapeutic outcomes. Human milk oligosaccharides (hMOS) are non-digestible carbohydrates known to exert health benefits in breastfed infants by preventing infection, maintaining immune homeostasis and nurturing healthy gut microbiota. These beneficial effects suggest the hypothesis that hMOS might have applications in COVID-19 as receptor decoys, immunomodulators, mucosal signaling agents, and prebiotics. This review summarizes hMOS biogenesis and classification, describes the possible mechanisms of action of hMOS upon different phases of SARS-CoV-2 infection, and discusses the challenges and opportunities of hMOS research for clinical applications in COVID-19.

## 1. Introduction

Approximately 219 million people worldwide have been affected by coronavirus disease 2019 (COVID-19) caused by severe acute respiratory syndrome-coronavirus-2 (SARS-CoV-2), and more than 5 million deaths were attributed to this disease as of November 4, 2021. SARS-CoV-2 is an enveloped, positive-sense single-stranded RNA virus, belonging to the group of Betacoronaviruses. SARS-CoV-2 is a zoonotic virus in origin that was first reported in humans in late 2019. The World Health Organization (WHO) declared the outbreak of SARS-CoV-2 in January of 2020, and a pandemic in March of 2020. Since then, many attempts have been made to launch COVID-19 vaccines as preventive measures against the ongoing global pandemic. Nonetheless, breakthrough infection has been increasingly reported due to the newly emerging SARS-CoV-2 variants (e.g., the Delta variant) [1]. SARS-CoV-2 mutations can have substantial effects on viral transmissibility and immune escape [2,3]. There are three variants of concern, two variants of interest, and nine other variants of SARS-CoV-2 currently posing a threat of human COVID-19 infection (source: www.ecdc.europa.eu/en/covid-19/variants-concern; last accessed on 4 November 2021). Until a “universal” pan-coronavirus vaccine to protect against all SAR-CoV-2 variants is discovered, therapeutic management will continue to utilize antiviral medication, immunomodulators, monoclonal antibodies, oxygen therapy and respiratory support as the mainstay treatment for patients with COVID-19. In addition, nutritional supplements with antiviral and immunomodulatory activities (i.e., vitamin C, vitamin D and zinc) have been used as adjunct therapies in the prevention and treatment of COVID-19 and its complications [4].

One of the most important complications is long-term sequelae in post-acute COVID-19 patients, known as long-COVID syndrome. Long-COVID is characterized by a range of persistent symptoms or development of sequelae beyond 4 weeks from the onset of acute symptoms of COVID-19 [5]. The incidence of long-COVID ranges from 10% to 72.5% of all COVID-19 patients [6,7]. Long-COVID involves multiple organs and manifests a wide range of symptoms including, but not limited to, fatigue (most common), dyspnea, palpitation, chest pain, muscle weakness, cognitive and executive function problems, renal function decrement, and thromboembolism [5,6,7,8]. The pathogenesis of this emerging syndrome remains under investigation. A range of possible mechanisms [8,9,10] include; (i) fibrotic changes after substantial tissue damage; (ii) central and peripheral nervous system and neuroendocrine involvement; (iii) chronic inflammation and miscommunication of immune response pathways, and/or; (iv) alterations in gut microbiota. Long-COVID syndrome can have severe impacts on health and socio-economic aspects in the post-COVID-19 era, but no definitive therapeutic intervention currently exists. One potential source of oral therapeutic molecules that attenuate many inflammatory and destructive aspects of pathobiology, including those caused by infections, inflammatory diseases, and gut dysbiosis [11,12], is human milk.

Human milk is the most suitable source of nutrition for newborns. Human milk contains macronutrients to support infant growth and development, and bioactive components that promote infant health [11]. Clinical and epidemiological data demonstrate that exclusive breastfeeding for the first six months of life can reduce the risk of mortality from infectious diseases [12]. Breastfed infants have lower risk of morbidity (the relative risk [RR] = 0.68; 95% confidence interval [CI] 0.60–0.77) and mortality (RR = 0.30; 95%CI 0.16–0.56) from lower respiratory tract infections compared to non-breastfed infants [12]. This beneficial effect of human milk is largely dependent on biologically active components, including growth factors, immunoglobulins, cytokines, antimicrobial peptides, and the non-digestible carbohydrates, human milk oligosaccharides (hMOS) [11]. HMOS comprise the third highest fraction of solid components in human milk. HMOS can exert various functions, i.e., anti-infective (against bacteria and viruses), signaling, anti-inflammatory/immunomodulatory, and prebiotic effects. To date, at least two structurally distinct hMOS, 2′FL and LNnT, have been approved by the US Food and Drug Administration (US FDA) and the European Food Safety Authority (EFSA) as novel ingredients for infant formulas [13].

The underlying question addressed by this review is whether the antiviral and immunomodulatory activities of hMOS could be useful in the prevention and treatment against COVID-19. Moreover, prebiotic effects of hMOS may be advantageous to many patients who suffer from gut dysbiosis during acute SARS-CoV-2 infection and post-COVID-19 syndrome. This review summarizes hMOS biogenesis and classification, describes the possible mechanisms of action of hMOS upon different phases of SARS-CoV-2 infection, and discusses the challenges and opportunities of hMOS research for clinical applications in COVID-19. Herein, the term hMOS (human Milk OligoSaccharide [singular or pleural]) refers to the mixture of human milk oligosaccharides, while specific names are used to indicate particular hMOS structures.

## 2. HMOS: A Brief Summary

HMOS are structurally diverse non-digestible glycans that are present in a distinct composition for each lactating mother. This individualized hMOS profile differs by maternal genetic status, physiology, ethnicity, and geography [14,15]. The diversity of patterns of hMOS in each mother remains throughout the course of lactation; however, the concentration of hMOS changes as lactation progresses [16,17]. HMOS are expressed in highest concentration in colostrum (9–22 g/L) and gradually decline in mature human milk; from 6–15 g/L at one-month to 4–6 g/L at six-months of lactation [17]. This information is clinically relevant to breastfeeding practice, and helps in designing future research related to hMOS diversity and functions.

HMOS are composed of five different monosaccharides (Figure 1a): glucose (Glc), galactose (Gal), N-acetylglucosamine (GlcNAc), fucose (Fuc), and N-acetylneuraminic acid (Neu5Ac; the prevalent form of sialic acid in humans) [18]. Lactose (Galβ1,4Glc) at the reducing end serves as the backbone of all hMOS (Figure 1b). Lactose can be extended through the addition of disaccharides lacto-N-biose (Galβ1,3GlcNAc) or N-acetyllactosamine (Galβ1,4GlcNAc). While the addition of lacto-N-biose stops the chain, that of N-acetyllactosamine allows further elongation. The branching of hMOS occurs with the addition of a β1,6 linkage between two disaccharides. The addition of fucose (fucosylation) onto lactose via α1,2 or α1,3 linkage results in trisaccharide 2′FL or 3FL, while lactose elongation by the addition of Neu5Ac (sialylation) via an α2,3 or α2,6 linkage produces the trisaccharides 3′SL or 6′SL [18]. Figure 1c illustrates some prevalent hMOS structures of human milk, some of whose biological activities will be discussed.

Differential hMOS fucosylation depends upon the presence or absence of two key enzymes, α1,2-fucosyltransferase encoded by the *Se* (Secretor; FUT2) gene, and α1,3/4-fucosyltransferase encoded by the *Le* (Lewis; FUT3) gene. Other fucosyltransferases, e.g., FUT4, 5, 6, 7, and 9, are capable of catalyzing the formation of α1,3 linkages. The activities of *Se* and *Le* gene loci determine the mother′s ability to secrete fucosylated hMOS into breastmilk and lead to a classification of four human milk groups as follows; group 1, Lewis-positive secretor (Se^+^Le^+^) produces α1,2/3/4-fucosylated hMOS; group 2, Lewis-negative secretor (Se^+^Le^−^) produces α1,2-fucosylated hMOS; group 3, Lewis-positive non-secretor (Se^−^Le^+^) produces α1,3/4-fucosylated hMOS; and group 4, Lewis-negative non-secretor (Se^−^Le^−^) produces α1,3-fucosylated hMOS independently of FUT2 and FUT3 expression [18,19]. The other α1,3 linkages are synthesized by fucosyltransferases expressed by FUT4, 5, 6, 7, or 9. Secretor status also determines the abundance of distinct glycans. For example, α1,2 fucosylated glycans (e.g., 2′FL and LNFP I) are enriched in breast milks of all secretor mothers, and all hMOS including α1,3/4-fucosylated glycans (i.e., 3FL, LNFP II, LDFH I/II) are present in the Lewis-positive secretor group. In contrast, the non-secretor group does not express α1,2-fucoslyltransferase, resulting in minimal α1,2 fucosylated glycans, with a concomitant increase in sialylated glycans in their breast milk.

HMOS sialylation depends on the activities of sialyltransferases. However, Bao et al. [20] found an unexpected association between hMOS sialylation and the secretor status by the Systems Biology approach. The reaction flux analysis of intermediate glycans revealed that the LSTb to DSLNT conversion rate was ~30% higher in the secretor group as compared to the non-secretor, whereas the conversion rate of LNT to LSTb was not affected [20]. Although the underlying mechanisms have yet to be defined, this indirect impact of the secretor status on hMOS sialylation suggested that the phenotype-specific reaction propensity could affect both abundance and diversity of hMOS in the four human milk groups [20].

The α1,2-fucoslyltransferase, encoded by the FUT2 secretor gene, is also expressed in multiple mucosal epithelial tissues to produce mucosal α1,2-fucosylated glycans, some of which are moieties found in mucosal histo-blood group antigens (HBGAs) [21,22]. Figure 1d showed different structures of HBGAs that serve as binding sites or attachment receptors for bacteria and viruses. Clinically, secretor infants have a higher risk of norovirus and rotavirus infections [23,24,25,26,27] and FUT2 variants could modify the infectious risk by altering HBGAs [26,27], thus supporting the roles of fucosylated receptors in host susceptibility to viral infections.

## 3. Potential Applications of hMOS in COVID-19

The beneficial effects of hMOS on preventing infection, maintaining homeostasis and nurturing healthy gut microbiota led to the hypothesis that hMOS might have applications in COVID-19. Herein, hMOS applications are reviewed and discussed with regard to four modes of action: as competitive receptor binding inhibitors (decoys), as anti-inflammatory and immunomodulation agents, as prebiotics, and as mucosal signaling agents (Figure 2).

### 3.1. HMOS as Receptor Decoys

The obligatory first step whereby SARS-CoV-2 virus initiates an infection is the binding of its spike protein to the human angiotensin converting enzyme 2 (ACE2) receptor on the mucosal cell surface of the host [28]. Human ACE2 receptor is heavily glycosylated; many of the glycans are fucosylated [29], and these glycosylated moities directly modulate the interaction between the viral spike protein and the host ACE2 receptor [29,30,31]. Several studies link A-type HBGA to this process [32,33,34,35,36,37].

Individuals with blood group A are more susceptible to SARS-CoV-2 infection [32,33]. In a meta-analysis, Golinelli et al. [32] reported that SARS-CoV-2 positive patients are more likely to have blood group A (pooled OR = 1.23, 95%CI 1.09–1.40) than group O (pooled OR = 0.77, 95%CI: 0.67–0.88). In another meta-analysis, Liu et al. [33] demonstrated COVID-19 patients with blood group A have an increased trend of mortality (OR = 1.25, 95%CI 1.02–1.52). Blood group A may also associate with COVID-19 disease severity and clinical outcomes [34,35]. Al-Youha et al. [34] conducted a retrospective study to determine the effects of blood group types in 3305 SARS-CoV-2-positive patients. Multivariate analysis revealed COVID-19 patients with blood group A associated with a higher risk of developing pneumonia compared with non-group A (adjusted OR = 1.32, 95%CI 1.02–1.72, *p* < 0.05) [34]. Mankelow et al. [35] reported that COVID-19 patients with blood group A had a higher risk of cardiovascular complications relative to blood group O (RR = 2.56, 95%CI 1.43–4.55; *p* = 0.0011). Interestingly, this study observed that blood group A patients who were secretors had a higher proportion of hospitalization than those who were non-secretors (20% vs. 8.1%, *p* = 0.019) [35].

Evidence from in vitro molecular studies strongly supports the interaction between SARS-CoV-2 and A-type HBGA [36,37]. Using a cellular model, Guillon et al. [36] observed that transfected SARS-CoV spike proteins were partially co-localized with A-type HBGA at the cell surface. Moreover, the interaction between SARS-CoV-2 spike protein and ACE2 receptor was diminished in the presence of anti-A antibodies [36]. The receptor-binding domain of SARS-CoV-2 spike protein preferentially bound to A-type 1 HBGA in a glycan microarray [37].

The molecular similarity of hMOS to host cell receptors allows hMOS to act as functional receptor decoys (competitive inhibitors) that block viral binding and entry. Neutral fucosylated and acidic sialylated hMOS demonstrate antiviral potential against several viruses, e.g., noroviruses [38,39,40,41,42], rotavirus [43,44,45], human immunodeficiency virus [46,47] and influenza virus [48,49]. An early definition of the hMOS receptor decoy mechanism in viruses is the interaction between hMOS and human norovirus, a leading cause of serious viral gastroenteritis globally [24,38,50]. Human norovirus requires binding to cell surface HBGAs for viral attachment to gastrointestinal epithelial cells. Fucosylated hMOS, such as 2′FL, 3FL, and LNFP I, structurally mimic HBGAs (Figure 1c,d) to prevent infections of multiple genotypes of human norovirus by competitively binding to P domain or capsid protein [39,40,41,42]. Table 1 shows selected studies on the receptor decoy effect of specific hMOS structures against viruses, including SARS-CoV-2. Taken together, these data support the hypothesis that fucosylated hMOS, especially 2′FL, may be able to inhibit SARS-CoV-2 viral entry via competitive binding inhibition.

### 3.2. HMOS as Anti-Inflammatory and Immunomodulatory Agents

The pathophysiology of COVID-19 has two main phases; viral replication dominates the early phase and immune dysregulation drives the later phase. Dysregulated immune responses in COVID-19 result in an imbalance between pro- and anti-inflammatory mediators [53]. The consequent loss of immune homeostasis manifests as: high proinflammatory cytokines (e.g., IL-6, IL-8, TNFα); profound lymphopenia; substantial immune cell defects (i.e., T cells, monocytes and dendritic cells) [53,54,55,56]; and impaired type I interferon response [57,58]. In severe cases, this complex immune dysregulation can lead to septic shock, acute respiratory distress syndrome, multi-organ failure, and death [59]. Therefore, the recommended current therapeutic management is to combine antiviral therapy with immunosuppressive drugs, i.e., dexamethasone (a corticosteroid), tocilizuman (an IL-6 receptor monoclonal antibody) or baricitinib (a selective JAK1/2 inhibitor) for hospitalized patients with moderate-to-severe COVID-19 in the later phase of infection (www.covid19treatmentguidelines.nih.gov; last accessed on 4 November 2021). While immunosuppressive therapies aim to mitigate the severe inflammatory response associated with COVID-19 infection, theoretical concerns have been suggested regarding delayed viral clearance and increased risk of opportunistic infections that could potentially be induced by potent anti-inflammtory agents.

Apart from the antiviral potential of hMOS, these milk glycans also exhibit anti-inflammatory and immunomodulatory activities (Figure 2b). Cumulative evidence strongly suggests that various hMOS can bind to glycan-binding proteins, or lectins, that are expressed on various cells to elicit mucosal and systemic immunomodulation [60,61]. HMOS exhibit anti-inflammatory and immunomodulatory effects through several mechanisms. HMOS can directly bind to surface lectins on immune cells to trigger changes in T cell proliferation, differentiation and anti-inflammatory cytokine production, e.g., IL-10 [62,63,64,65,66,67], and dendritic cell maturation [67,68,69,70]. Table 2 summarizes the vast immunomodulatory effects of specific hMOS in various cell types.

The association between hMOS and type I interferon has rarely been evaluated. Type I interferon (mainly IFNα/β) is part of an innate immune mechanism against viral infection. Type I interferon activates IFNα receptor1/2, triggers STAT1/2 and IFN-regulatory factor 9 (IRF9) and induces IFN-stimulatory genes (ISGs) which establish the antiviral state in infected cells and their neighboring cells to suppress viral replication [71,72,73]. Human plasmacytoid dendritic cells, a subset of mature dendritic cells, are recognized as the viral specialists that massively release type I interferon upon viral stimulation [74]. Melendi et al. [75] report that human milk triggers type I interferon production in infants infected with influenza virus. This antiviral induction by human milk could extend to other respiratory viruses, e.g., respiratory syncytial virus, parainfluenza virus, human metapneumovirus, and perhaps SARS-CoV-2 [76]. HMOS are major bioactive compounds in human milk and exert immunomodulatory effects through dendritic cells [67,68,69,70]; we hypothesize that hMOS also trigger type I interferon production in infected cells, or could induce dendritic cell maturation into the plasmacytoid subset, which would promote an antiviral state. However, data supporting this hypothesis are currently not available.

**Table 2 biomedicines-10-00346-t002:** Evidence supporting the immunomodulatory roles of hMOS.

HMOS	Targets/Models	Immunomodulatory Effects	References
Acidic	Cord blood-derived mononuclear cells; human	Induce Th-1 cytokine IFN-γ and regulatory cytokine IL-10, but not Th-2 cytokines IL-13, IL-4 and IL-12 causing Th-1 shift	[63]
	Allergen-specific CD4+ T-cells; human	Significantly suppress Th-2 cytokine IL-4 and slightly reduce IL-13	[63]
Mixture (human milk isolates)	Dendritic cells; human	Induce dendritic cell maturation via TLR4/DC-SIGN interaction, releasing IL-10 and promote regulatory T cell differentiation from T naïve cells	[67]
Mixture (2′FL, LNnT, 3′SL, 6′SL, free sialic acid)	PBMCs, pig	Increase numbers of peripheral blood NK cells and effector memory T cells	[77]
2′FL	2′FL containing formula fed healthy infants	Decrease plasma levels of proinflammatory cytokines IL-1ra, IL-1α, IL-1β, IL-6 and TNFα relative to control formula-fed infants	[78]
	PBMCs; pigs	Suppress proliferation of PBMCs and CD4 + T cells	[62]
	Dendritic cells; human	Induce IFNγ and IL-10 secretion by CD4+ T cells	[68]
	Dendritic cells; human	Bind specifically to DC-SIGN receptor (IC50 of ~1 mM), influencing dendritic cell functions	[79]
LDFT	Platelet; human	Inhibit platelet-induced inflammatory processes by suppressing release of proinflammatory proteins, i.e., RANTES, sCD40L	[80]
LNFPIII	Peritoneal macrophages; mice	Activate macrophages independent of IL-4/IL-13 cytokines, and induce IL-10 secretion; Adoptive transfer of LNFPIII-stimulated macrophages induced IL-10 and IL-13 expression in recipient naïve T cells, and activated NK cells	[65,66]
	Peritoneal macrophages; SCID mice	Activate and expand suppressor F4/80 + Gr1+ macrophage population in a T cell-independent mechanism	[81]
	Spleen cells; mice	Induce IL-10 production and B cell proliferation	[64]
	PBMCs; pigs	Induce IL-10 production and inhibit T cell proliferation	[62]
	Dendritic cells; mice	Induce dendritic cell maturation	[69]

Abbreviations: 2′FL, 2′-fucosyllactose; 3′SL, sialyl(α2,3) lactose; 6′SL, 6′-sialyllactose; Ag, antigen; GLs, galactosyllactoses, hMOS, Human milk oligosaccharide(s); IL-1ra, interleukin-1 receptor antagonist; LNFPIII, lacto-N-fucopentaose III; PBMCs, peripheral blood mononuclear cells; SCID, severe combined immunodeficiency; TLR4, toll-like receptor 4.

### 3.3. HMOS as Mucosal Signaling Agents

SARS-CoV-2 spike protein can bind to and activate toll-like receptor 4 (TLR4) signaling to mediate respiratory mucosal inflammation and acute lung injury [82,83,84,85]. Zhao et al. showed that the receptor-binding domain (RBD) of SARS-CoV-2 spike protein induce IL-1β and IL-6 expression in THP-1 immortalized human monocytes, and IL-1β expression in primary macrophages derived from mouse peritoneum and bone marrow [82]. IL-1β induction by the spike protein was diminished by resatorvid, a selective TLR4 inhibitor, and likewise, in TLR4-deficient mice. Since CD14 is a co-receptor of TLR4, blocking CD14 with anti-CD14 antibody could also suppress IL-1β induction by the SARS-CoV-2 spike protein [82]. Interestingly, IL-1β induction by SARS-CoV-2 spike protein was comparable to lipopolysaccharide (LPS) treatment [82]. While the detailed molecular mechanisms have yet to be elucidated, antagonizing TLR4 signaling has been recognized as a new therapeutic approach to mitigate the severity of COVID-19 pneumonia [83,84,85].

HMOS are known to modulate mucosal signaling cascades including TLR4 (Table 3). HMOS alter gene expression in the intestinal epithelial cells which promote intestinal cell maturation, mucosal barrier function, tissue repair and tight junction integrity while attenuating LPS-induced inflammation through suppressing TLR4 signaling [86,87,88,89,90]. HMOS isolated from colostrum could directly modulate mucosal inflammatory signaling of immature human intestinal tissue by attenuating the expression of proinflammatory cytokines (i.e., IL-1β, IL-6, IL-8, and TNFα) [86,91]. HMOS also modulate the development of the intestinal epithelial glycocalyx that supports microbial colonization and mucosal barrier function. 2′FL and 3FL enhance intestinal glycocalyx development in a structure-dependent manner [92]. Intestinal glycocalyx development induced by 2′FL could prevent the adhesion of pathogenic *E. coli* [93]. In addition, 2′FL suppresses pathogenic *E. coli* and *Campylobactor jejuni*-induced mucosal inflammatory signaling of intestinal epithelial cells. [87,94]. HMOS, including 2′FL and 6′SL, directly bind to TLR4 and suppress TLR4-mediated NF-κB signaling to prevent necrotizing enterocolitis, a major cause of mortality in preterm infants [95]. Furthermore, 2′FL can attenuate the expression of CD14, the co-receptor of TLR4 [87], thus supporting its antagonistic effects against TLR4-mediated mucosal inflammation.

To date, several phase I/II clinical trials have been conducted to evaluate the efficacy of TLR4 modulators in COVID-19 (Clinicaltrials.gov NCT02735707, NCT04401475, NCT04479202). We therefore anticipate that hMOS, either as mixtures or as individual molecules, could prove to be useful as a mucosal signaling agent with TLR4 inhibitory activity to protect against respiratory mucosal inflammation and acute lung injury in patients with severe COVID-19.

### 3.4. HMOS as Prebiotics to Mitigate Gut Dysbiosis

Gut microbiota can provide health benefits by controlling intestinal homeostasis and regulating local and systemic immune responses [100,101,102]. Biological effects of gut microbiota have been explained by the crosstalk of small molecules and metabolites between intestinal mucosa and bacteria [103,104,105,106,107]. Microbiota-derived short chain fatty acids (SCFAs), including acetate, butyrate and proprionate, are produced by the fermentation of non-digestible carbohydrates, and especially oligosaccharides, by healthy gut microbiota. SCFAs provide energy to intestinal epithelial cells, impact intestinal homeostasis, and exhibit anti-inflammatory and immunomodulatory effects [103,104]. SCFAs, particularly butyrate, also play key roles in the microbiota-gut-brain axis and regulating central nervous system functions [105,106]. Conversely, gut dysbiosis allows an increase in pathogens, which generate toxic products, especially lipopolysaccharides (LPS), that aggravate mucosal and systemic inflammation [107].

COVID-19 patients can exhibit dysbiosis from the onset of illness through viral clearance and this can persist even after disease resolution [9,10,108,109]. Gut dysbiosis is characterized by the imbalance between beneficial mutualist bacteria (i.e., *Faecalibacterium prausnitzii*, *Roseburia, Lachnospiraceae, Eubacterium, Bifidobacterium* and *Lactobacillus*) and opportunistic pathogens (such as *Streptococcus, Rothia*, *Veillonella*, *Clostidium*, and *Actinomyces*) [9,10,108,109]. Zuo et al. [108] evaluated fecal microbiota from 15 patients with COVID-19 and found that a baseline abundance of *Clostridium ramosum* and *Clostridium hathewayi*, and a low abundance *Faecalibacterium prausnitzii* are associated with COVID-19 severity. Gu et al. [109] demonstrated that opportunistic pathogens, i.e., *Streptococcus*, *Rothia*, *Veillonella*, *Erysipelatoclostridium* and *Actinomyces*, were relatively enriched in stool samples collected from 30 COVID-19 patients; this altered gut microbiota had a positive correlation with the inflammatory marker C-reactive protein (CRP). Of note, *Streptococcus* and *Rothia* are associated with increased risk of secondary bacterial pneumonia in patients with avian influenza virus infection [110]. *Rothia* may contribute to the pathogenesis of pneumonia in immunocompromised patients [109,111], while *Faecalibacterium prausnitzii* has anti-inflammatory properties [112].

There are significant associations between gut dysbiosis, inflammatory cytokines and other inflammatory proteins in COVID-19 disease severity. Yeoh et al. [9] reported that commensal bacteria including *Faecalibacterium prausnitzii*, *Eubacterium rectale*, and *Bifidobacterium* species were low in the stool samples obtained from COVID-19 patients (100 patients vs. 78 non-COVID-19 controls) during hospitalization and these low levels persisted for up to 30 days after viral clearance [9]. Elevated levels of inflammatory cytokines (e.g., TNFα, CXCL10), and inflammatory markers of tissue damage (such as C-reactive protein (CRP), lactate dehydrogenase, aspartate aminotransferase, gamma-glutamyl transferase, and N-terminal-pro-brain natriuretic peptide) were negatively correlated with gut dysbiosis in patients with severe COVID-19 [9]; this suggests that altered microbiota communities could play roles in regulating immune responses and disease severity in COVID-19. Chen et al. [10] evaluated alterations of gut microbiota in 30 patients with COVID-19 at acute phase (from the onset of illness to viral clearance), convalescence (from viral clearance to two weeks after hospital discharge) and postconvalescence (six months after hospital discharge). The gut microbiota richness was remarkably reduced in the acute phase (Chao1 index of 217) and did not recover during convalescence and postconvalescence (Chao1 indices of 241 and 259, respectively), as compared to the uninfected healthy controls (*n* = 30; Chao1 index of 432). Patients with depressed gut microbiota richness during postconvalescence had higher degrees of disease severity and plasma CRP levels in the acute phase [10]. These data suggest that therapeutic interventions that mitigate gut dysbiosis and restore mutualist immunoregulatory bacteria are promising approaches toward treating and preventing COVID-19 disease progression and long-term complications.

We conclude that prebiotic oral supplements have high potential for treating COVID infections and their sequellae, as they may be efficacious, would complement any other treatments, and perhaps could be the most attractive mode of therapeutic applications in COVID-19 patients (Figure 2d). HMOS strongly effect the assembly of a healthy gut microbiota ecosystem, serving as the selective growth substrate for beneficial mutualists, including various *Bifidobacteria*. HMOS fermentation by the healthy gut microbiota, in turn, produce SCFAs which can exert beneficial effects on intestinal homeostasis, immunomodulation, and gut-brain axis [103,104,105,106].

## 4. Challenges and Opportunities

Some sialylated and fucosylated glycans [37,52] bind to the SARS-CoV-2 receptor binding domain (RBD) region, posing the possibility that hMOS containing such glycan motifs could block viral entry and infectivity. While hMOS have been recognized for potential application in COVID-19 treatment, direct evidence of anti-SARS-CoV-2 activity by hMOS is not evident in the literature. To obtain such data, one strategy is to utilize cell-based high-throughput antiviral screening, which was recently adapted for small molecules and natural compounds [113,114,115,116]. Positive results from these designs would support further development of hMOS as the antiviral agent for COVID-19.

Determining the best route of hMOS administration to combat COVID-19 infection requires new avenues of research. For example, the mechanisms whereby hMOS are absorbed from gut into bloodstream remain unknown. Future research should address the physicochemical and structural properties of various oligosaccharides as determinants of mucosal permeability or transport. Of course, the natural route of hMOS administration is oral, and hMOS are resistant to digestive enzymes in human gastrointestinal tract. Of the total ingested hMOS, 4% and 50% are excreted through urination and defecation, while 45% are fermented by gut microbiota [117,118,119,120]. The remaining 1% of ingested hMOS are absorbed from gastrointestinal tract into systemic circulation, resulting in plasma concentration of 1–10 mg/L [117,118,119,120]. These data support the use of hMOS as a prebiotic supplement, aiming to restore gut microbiota, and may support systemic effects of hMOS to mitigate hyperinflammatory responses during acute infection as well as chronic inflammation during long-term recovery of COVID-19. It is unknown whether the oral administration of hMOS, or lung deposition of micro-droplets generated following liquid swallowing, could achieve therapeutic levels of hMOS in the human respiratory tract.

To improve hMOS delivery to lung parenchyma, aerosol inhalation might be an alternate route (Figure 3a). From the anatomical viewpoint, aerosol inhalation would deliver hMOS to exert competitive inhibition against SARS-CoV-2 entry and promote mucosal barrier function covering the entire respiratory tract. While the safety of hMOS by aerosol inhalation therapy has not been reported, the inhaled dry power alginate oligosaccharide derived from seaweed has been undergoing phase I/II clinical trials for the safety and efficacy in patients with cystic fibrosis [121]. The number and proportions of patients with adverse events, e.g., nasopharyngitis, cough, and pulmonary exacerbation were similar between the inhaled oligosaccharide and the placebo arms during the 28-day period [121]. The safety and efficacy of oligosaccharides as part of inhalation therapy in pulmonary diseases [121,122] suggest probable safety of aerosolized hMOS treatments in preclinical studies and early phase clinical trials for COVID-19.

HMOS intravenous administration is theoretically a simple route to evaluate and harness the fully potential benefits of hMOS on the systemic immunomodulation (Figure 3b). Once the safety profile is established in animals, phase I clinical trials could be conducted in healthy individuals for safety evaluation. This route of administration has several drawbacks, including the intravenous route being more invasive and inconvenient (or inaccessible) for non-hospitalized patients. Note that 1% of the ingested hMOS is absorbed and 50% remains in the gastrointestinal tract until defecation. However, we do not know the mechanisms whereby hMOS are absorbed from gut into bloodstream. Structure-function studies to determine the relationship between physicochemical and structural properties of hMOS in relation to mucosal permeability or transport are needed. The results of such studies may guide creation of suitable formulations or modification that improve hMOS bioavailability via the oral route (Figure 3c,d).

## 5. Conclusions

HMOS hold great promise for the prevention and treatment of COVID-19 by four modes of action, including competitive inhibition (receptor decoy), anti-inflammatory and other immunomodulation, mucosal signaling with TLR4 inhibition, and prebiotic action. Further investigations focusing on anti-SARS-CoV-2 effects, the routes of administration and the oral bioavailability of hMOS are warranted for clinical applications of hMOS as a new therapeutic/nutraceutical agent for COVID-19.

## Figures and Tables

**Figure 1 biomedicines-10-00346-f001:**
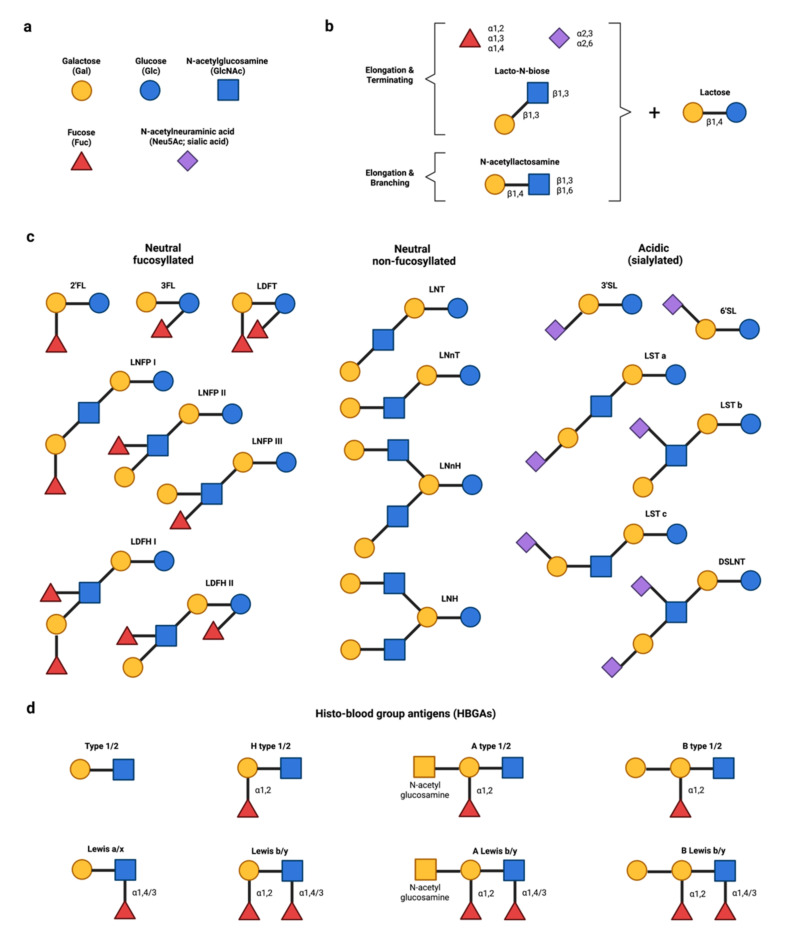
Structural diversity of hMOS and their analogous human blood group antigens (HBGAs). (**a**) HMOS are constituted from five different monosaccharides. (**b**) All hMOS have lactose at the reducing end as the building block. The structural diversity of hMOS is due to combinations of elongation with N-acetyllactosamine, Lacto-*N*-biose, fucose and/or sialic acid. (**c**) Selected hMOS structures commonly found in human milk, some of which are further discussed in this review. (**d**) HBGAs serve as attachment receptors for several viruses. Fucosylated hMOS (e.g., 2′FL) can structurally mimic receptors, thereby acting as decoys (competitive inhibitors) of HBGAs. Abbreviations: 2′FL, 2′-fucosyllactose; 3FL, 3-fucosyllactose; 3′SL, 3′-sialyllactose; 6′SL, 6′-sialyllactose; hMOS, Human milk oligosaccharides; DSLNT, Disialyllacto-*N*-tetraose; LDFH, Lacto-*N*-difucohexaose; LDFT, Lactodifucotetraose; LNFP, Lacto-*N*-fucopentaose; LNH, Lacto-*N*-hexaose; LNnH, Lacto-*N*-neohexaose; LNT, Lacto-*N*-tetraose; LNnT, Lacto-*N*-neotetraose; LST, Sialyllacto-*N*-tetraose.

**Figure 2 biomedicines-10-00346-f002:**
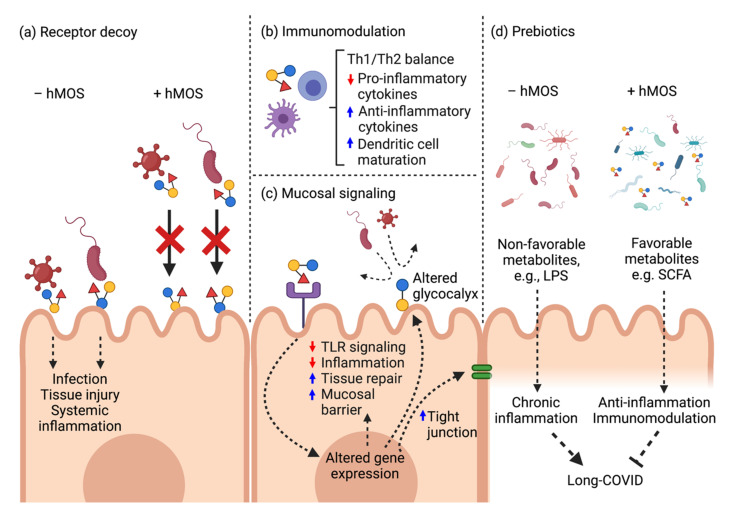
Potential modes of action of hMOS against SARS-CoV-2. (**a**) HMOS molecular structures are analogous to HBGAs and act as receptor decoys to block viral entry. (**b**) HMOS induce local defense and immumomodulation. (**c**) HMOS attenuate TLR4-mediated signaling pathways to maintain mucosal homeostasis. (**d**) HMOS mitigate gut dysbiosis and restore healthy gut microbiota in long-COVID.

**Figure 3 biomedicines-10-00346-f003:**
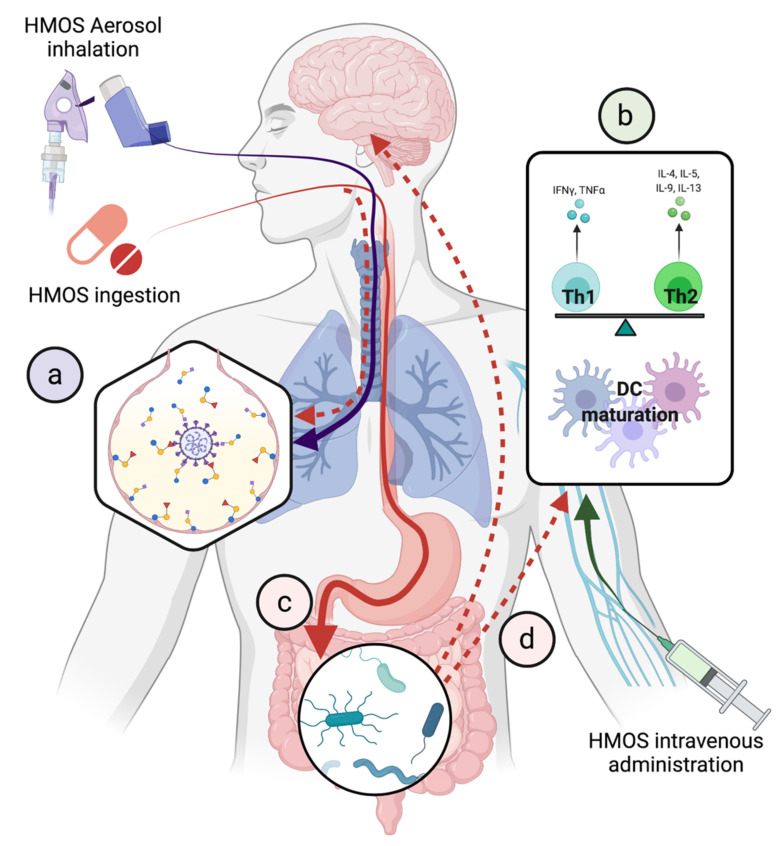
Proposed therapeutic applications of hMOS in COVID-19. (**a**) Aerosolized inhalation for prevention and/or treatment. (**b**) Intravenous administration for systemic anti-inflammation and immunomodulation. (**c**) Oral administration for prebiotic effects to restore gut dysbiosis which may mitigate chronic inflammation and neurological symptoms via gut-brain axis in long-COVID through (**d**) systemic absorption of ingested hMOS.

**Table 1 biomedicines-10-00346-t001:** Evidence supporting antiviral effects of hMOS via the receptor decoy mechanism.

Virus	Glycans	Mechanisms and Functions	References
Coronavirus			
MERS-CoV	α2,3-sialytated glycans	Mimic sialylated attachment receptor; bind to MERS-CoV spike protein; may inhibit MER-CoV spike-DPP4 interaction and block viral entry into host cells	[51]
SARS-CoV	A-type HBGA	Co-localized with the transfected SARS-CoV spike protein	[36]
SARS-CoV-2	A-type HBGA	Bind to SARS-CoV-2 RBD of spike protein; may modulate viral entry	[37]
	Sialylated glycans	Bind to SARS-CoV-2; may modulate viral entry	[52]
HIV	Le^x^	Block DC-SIGN on dendritic cells to prevent HIV gp120 envelop protein interaction; inhibit DC-SIGN-mediated transfer of HIV-1 to CD4 + T lymphocyte	[46,47]
Influenza virus Avian influenza	3′SL, 6′SL	Mimic sialylated host cell receptor; block Influenza virus envelop protein, haemagglutinin, interacting with host cells	[48,49]
Novovirus	2′FL, 3FL, LNFP I	Mimic HBGAs; block human novovirus P domain or capsid protein interacting with blood group–active mucin-type*O*-glycans on host cell surface	[39,40,41,42]
Rotavirus			
G1 [8], G2P [4]	2′FL, 3′SL, 6′SL	Inhibit viral infectivity	[44]
P [8]	LNB	Mimic secretory H type-1 antigen; bind Rotavirus VP8* and inhibit viral infectivity	[45]
RV OSU	3′SL, 6′SL	Inhibit viral cellular binding and infectivity	[43]

Abbreviations: 2′FL, 2′-fucosyllactose; 3′SL, 3′-sialyllactose; 6′SL, 6′-sialyllactose; DC-SIGN, dendritic cell-specific ICAM3-grabbing non-integrin; DPP4, dipeptidyl peptidase 4; FECV, Feline enteric coronavirus; HBGA, Histo-blood group antigen; HIV, human immunodeficiency virus; hMOS, Human milk oligosaccharides; LDFH I, Lacto-N-difucohexaose I; Le^x^, Lewis X; LNB, Lacto-N-biose; LNFP I, Lacto-N-fucopentaose I; RBD, Receptor-binding domain.

**Table 3 biomedicines-10-00346-t003:** Evidence supporting roles of hMOS in mucosal signaling and epithelial protection.

HMOS	Targets/Models	Mucosal Signaling Effects	References
Acidic	Intestine; NEC model rat	Attenuate TLR4/NF-κB/NLRP3-mediated inflammation and suppress inflammatory signals of IL-1β, IL-6, TNFα to prevent NEC development	[96]
Mixture (human colostrum isolates)	Immature intestinal tissue; aborted fetuses	Attenuate pathogen-associated molecular pattern-stimulated IL-1β, IL-6, IL-8, MCP-1 expression while promoting MIP-1-δ, MIP-1-β, TIMP-2 and PDGF the mediators of tissue repair	[86]
Mixture (human milk isolates)	Intestinal epithelial cells in vitro; human	Suppress TNFα and IL-1β induced inflammtory signals of IL-8, MIP-3α and MCP-1	[91]
	Intestinal epithelial cells in vitro; human	Enhance epithelial differentiation and promote alkaline phosphatase activity	[88]
2′FL	Intestinal cells; human, mice, pigs	Attenuate CD14 expression and suppress LPS-induced IL-8 production in ETEC exposed intestinal cells	[87]
	Intestinal epithelial cells in vitro; human	Suppress Campylobactor jejuni-induced mucosal inflammatory signals of IL-1β, IL-8, MIP-2	[94]
	Intestinal epithelial cells in vitro; human, mice	Suppress TLR4 expression and TLR4-mediated NF-κB signaling to prevent intestinal inflammation and NEC development	[95]
	Intestinal epithelial cells in vitro; human	Selectively inhibit CCL20 release from Ag-IgE complex stimulated intestinal cells in a PPARγ independent manner	[97]
	Intestinal epithelial cells in vitro; human	Induce upregulation of DEFB1 and ZO-1 genes under the peristalsis-mimic shear force and promote tight junction formation	[98]
	Goblet cells in vitro; human	Induce upregulations of mucus associated genes TFF3 and CHST5 and promote the mucus barrier function	[90]
	Intestinal epithelial cells in vitro; human	Modulate glycosylation genes of galectin and downregulate ICAM-1 to prevent pathogen adhesion	[93]
3′SL	Intestine; IL-10(-/-) colitis mice	Promote colitis severity and modulated mucosal immunity by stimulating CD11c + dendritic cells through TLR4 pathway	[70]
	Intestinal epithelial cells in vitro; human	Induce upregulation of DEFB1 and ZO-1 genes under the peristalsis-mimic shear force and promote tight junction formation	[98]
6′SL	Intestinal epithelial cells in vitro	Inhibit chemokine (IL-8 and CCL20) release from Ag-IgE complex stimulated intestinal cells	[97]
	Intestine; human, mice, pigs	Suppress TLR4 expression and TLR4 signaling to prevent NEC development	[95]
DSLNT	Intestine; NEC model rat	Attenuate mucosal inflammation by a selectin-independent process to prevent NEC development	[99]

Abbreviations: DEFB1, defensin β-1; DSLNT, disialyllacto-N-tetraose; ETEC, enterotoxigenic E. coli; MCP1, monocyte chemoattractant protein 1; MIP, macrophage inflammatory protein; NEC, necrotizing enterocolitis; NF-κB, nuclear transcription factor-κB; PDGF, platelet-derived growth factor; TFF3, trefoil factor 3; TIMP-2, tissue inhibitor of metalloproteinase-2; TJP-1, Tight junction protein-1; ZO-1, zonula occludens-1.

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
