# Peer review of "Human Milk Oligosaccharides: Potential Applications in COVID-19"

_biomedicines, 2022, doi:10.3390/biomedicines10020346_

Round 1

Reviewer 1 Report

The topic of this paper is very interesting and it is very well written. I only have one observation to make:

Introduction: line 67 "A possible source..." It is a little bit unrelated, there is nothing about the incidence or mortality of covid in newborns. Why should we use human milk as a source of therapeutic molecules? It is too generic...

Author Response

Response to the Reviewer’s comments

MS: biomedicines-1572207

Reviewer#1

The topic of this paper is very interesting and it is very well written. I only have one observation to make:

Introduction: line 67 "A possible source..." It is a little bit unrelated, there is nothing about the incidence or mortality of covid in newborns. Why should we use human milk as a source of therapeutic molecules? It is too generic...

RESPONSE: Thank you for your kind review of our manuscript, and for pointing out this weakness. We revised line 67 as follows:

Introduction; page 2, lines 67-69:

“… Long-COVID syndrome can have severe impacts on health and socio-economic aspects in the post-COVID-19 era, but no definitive therapeutic intervention currently exists. One potential source of oral therapeutic molecules that attenuate many inflammatory and destructive aspects of pathobiology, including those caused by infections, inflammatory diseases, and gut dysbiosis [11,12], is human milk.”

Thank you for your efforts toward improving this review.

Reviewer 2 Report

Dear Editor and Authors,

I enjoyed reading the review. The content is impressive and it is a comprehensive review about the potential role of human milk oligosaccharides in protection against SARS-COV2 infection. Minor changes in language and some typos are highlighted in the attached PDF document. Thanks so much 

Author Response

Response to the Reviewer’s comments

MS: biomedicines-1572207

Reviewer#2

I enjoyed reading the review. The content is impressive and it is a comprehensive review about the potential role of human milk oligosaccharides in protection against SARS-COV2 infection. Minor changes in language and some typos are highlighted in the attached PDF document. Thanks so much.

RESPONSE: Thank you for your constructive comments and helpful suggestions toward improving the quality of our manuscript. All changes are highlighted in the red.

Comments

  1. Page 4, line 148: From “system biology approach” to “Systems Biology approach”

RESPONSE: This term has been corrected.

  1. Page 5, lines 194-196: Doublecheck the Secretor Fx from earlier

RESPONSE: We doublechecked the indicated paragraph in relation to the Secretor function mentioned earlier. The relationship between secretor gene induction and blood group type is complex, due to differences in mucosal control of expression of the FUT2 gene. We confirm that information in the indicated statement is consistent with our understanding of these systems.

  1. Page 6, line 204 (now 208): From “Molcular” to “Molecular”

RESPONSE: Thank you for finding this typographical error.

  1. Page 7, lines 225-229:

RESPONSE: Thank you for directing us to this sentence, which was too long and hard to read. Accordingly, the sentence was rewritten as follows:

“… the later phase. Dysregulated immune responses in COVID-19 result in an imbalance between pro- and anti-inflammatory mediators [53]. The consequent loss of immune homeostasis manifests as: high proinflammatory cytokines (e.g., IL-6, IL-8, TNFa); profound lymphopenia; substantial immune cell defects (i.e., T cells, monocytes, and dendritic cells) [53–56]; and impaired type I interferon response [57,58]. In severe …”

  1. Page 8, lines 281-282: Change HMOS to hMOS to remain consistent with the rest of the paper.

RESPONSE: In this review, when hMOS was the first word in a sentence, we decided to use this abbreviation with the capital letter H. In manuscripts of others, we have seen both examples of using capitalization of acronyms when they begin sentences, and retention of the lower case. When faced with the dilemma of two opposing rules, we went with capitalization. However, if in this light the Reviewer still prefers that the hMOS be used, we would be pleased to modify the manuscript accordingy.

  1. Page 11, lines 354-356 (now 359-361) : From “Chow1” to “Chao1”

RESPONSE: Thank you for pointing out this error.

  1. Page 13, line 415-419: Very important line - should be placed in paragraph starting line 379.

RESPONSE: The introduction to the paragraph (formerly 378, now 384-389) now reads:

“Determining the best route of hMOS administration to combat COVID-19 infection requires new avenues of research. For example, the mechanisms whereby hMOS are absorbed from gut into bloodstream remain unknown. Future research should address the physicochemical and structural properties of various oligosaccharides as determinants of mucosal permeability or transport. Of course, the natural route of hMOS administration is oral, and hMOS are resistant to digestive enzymes in human gastrointestinal tract.”

The end of this section, formerly 415-419, now 426-431, was modified to:

However, we do not know the mechanisms whereby hMOS are absorbed from gut into bloodstream. Structure-function studies to determine the relationship between physicochemical and structural properties of hMOS in relation to mucosal permeability or transport are needed. The results of such studies may guide creation of suitable formulations or modification that improve hMOS bioavailablity via the oral route (Figure 3c,d).

We thank this Reviewer for the thoughtful suggestions toward improving this manuscript.